# Effect of Interaction between Pt and Different Crystalline Phases of TiO$_2$ on Benzene Oxidation

**Hang Cheng** [1,2], **Jiangliang Hu** [1,2,3,*], **Dongxia Wu** [1,2], **Weiren Bao** [1,2,3], **Changming Hou** [1,2], **Xianyan Lv** [1,2], **Liping Chang** [1,2] and **Jiancheng Wang** [2,4]

1   College of Chemical Engineering and Technology, Taiyuan University of Technology, Taiyuan 030024, China; ch15552500238@163.com (H.C.); dongxiawu@yeah.net (D.W.); baoweiren@tyut.edu.cn (W.B.); houchangming1010@link.tyut.edu.cn (C.H.); lvxianyan1998@163.com (X.L.); lpchang@tyut.edu.cn (L.C.)
2   State Key Laboratory of Clean and Efficient Coal Utilization, Taiyuan University of Technology, Taiyuan 030024, China
3   New Materials and Chemical Research Institute, Shanxi Zhejiang University, Taiyuan 030024, China
4   College of Environmental Science and Engineering, Taiyuan University of Technology, Taiyuan 030024, China
*   Correspondence: hujiangliang@tyut.edu.cn

**Abstract:** To evaluate the effects of different TiO$_2$ crystalline phases on the catalytic oxidation performance of benzene on Pt-loaded TiO$_2$ catalysts, physicochemical examinations were conducted using several spectroscopic and analytical techniques. Obvious effects on the valence state and morphology of Pt were exhibited by different crystalline phases. The rutile phase favored the formation of specific Pt(111) crystal faces, which enhanced the amount of surface-active oxygen species. Moreover, the àPt-O-Ti species was formed between Pt$^{4+}$ and Ti at the edge of the Pt nanoparticles, promoting both electron flow and the transfer of reactive oxygen species, thus accounting for catalytic activity.

**Keywords:** TiO$_2$ crystalline phase; strong metal–support interaction; catalytic oxidation of benzene; active oxygen species





## 1. Introduction

Volatile organic compounds (VOCs) are the main atmospheric pollutants, and they mainly include non-methane alkanes, aromatic hydrocarbons, olefins, halogenated hydrocarbons, esters, aldehydes, ketones, etc. On the one hand, these compounds act indirectly as the main precursors of PM$_{2.5}$ (fine particulate matter) and O$_3$, inducing haze and producing photochemical smog. On the other hand, they act directly as toxic substances that are detrimental to the environment and human health [1–4]. Aromatic hydrocarbons (e.g., benzene) are typical examples of VOCs, and they are characterized by strong toxicity and carcinogenicity. Most are introduced from industrial emissions and automobile exhaust. Currently, several control technologies exist for VOCs, among which catalytic oxidation is prominent due to its wide application range, the extensive treatment of exhaust gas concentrations, and the non-generation of secondary pollutants; hence, it is recognized as one of the most effective methods for controlling VOCs.

Catalyst design is key in the catalytic oxidation method. For oxidation reactions, the main catalyst types are supported precious metal catalysts and non-precious metal catalysts. Supported noble metal catalysts have been the focus of extensive investigations because of their excellent catalytic activity and stability at low temperatures. In many cases involving TiO$_2$ as the support, the catalytic activity of the employed catalysts is effectively improved due to the strong interactions (SMSIs), chemical force effects, and good electron mobility between TiO$_2$ and noble metals. It is worth noting that most research studies on Pt/TiO$_2$ catalysts mainly focus on the strong interaction of metal supports caused by high-temperature reduction with a reducing atmosphere [5,6]. When applying the high-temperature reduction of TiO$_2$, strong hydrogen spillage results in the formation of an

amorphous $TiO_X$, and the electron interaction formed between Pt and $TiO_X$ ($Pt^{\delta+}/Pt^0 \leftrightarrow Ti^{3+}/Ti^{4+}$) can significantly affect the oxidation performance of the catalyst [7].

Recently, a benzene oxidation study with Pt as the active metal accompanied by the reducible oxide $TiO_2$ was carried out under certain conditions. According to the research of Liu et al. [8], $Pt/TiO_2$ microspheres with sea urchin-like structures were prepared via a simple hydrothermal method, and its high specific surface area and pore structure were more beneficial to the dispersion of Pt. However, their report mainly investigated the morphology of $Pt/TiO_2$ and the influence of treatment conditions on the oxidation performance of benzene, and they did not discuss the influence of $TiO_2$ morphology on the Pt state substantially. In addition, the oxidative activity of the catalyst also depends on the interface between the active metal and the catalyst's support. Geo et al. [9] studied the relationship between the SMSI effect on Pt particle size and valence state, and highly dispersed small metal Pt particles were more beneficial to the oxidation of CO, while catalysts containing $PtO_X$ had relatively poor activity. However, their research focused on the role of the active metal Pt rather than the $TiO_2$ support, and the nature of the catalyst support is also an important factor affecting catalyst activity.

Lastly, the $TiO_2$ crystalline phase is also a very important factor affecting the Pt state. However, the interaction relationship between $TiO_2$ and Pt in different crystalline phases has been reported relatively rarely, and $TiO_2$ mainly exists in three crystal forms, such as anatase, brookite, and rutile, with each crystal form exhibiting different physicochemical properties. It is well known that most catalytic reactions in catalysts occur on the surface of the catalyst or at the interface between the active materials and support materials [10]. Therefore, $TiO_2$-supported catalysts with different crystal structures exhibit different catalytic activities. It is worth noting that brookite displays metastability, which limits its application, and this is attributed to the relatively unstable crystal structure of its ore. Thus, the applications of anatase and rutile $TiO_2$ as catalysts were at the center of a large number of studies. It was reported that the oxidation properties of noble metal catalysts were largely dependent on the size of the active metal [11,12]. Using environmental transmission microscopy and density functional theory calculations, Zhang et al. [13] investigated the different arrangement behavior of Pt in anatase $TiO_2$ and rutile $TiO_2$, and they observed that while the rutile $TiO_2$ facilitated the reshaping of the Pt NP into two-dimensional plane geometries, which is beneficial for methane combustion, the anatase surface facilitated the redispersion of the Pt NP into single atoms during the same calcination process, which promotes the selective hydrogenation of phenylacetylene. Jiang et al. [14] investigated the $TiO_2$ of the three crystal phases of $Pt/$rutile $TiO_2$ (rutile phase $TiO_2$ synthesized via the hydrothermal method), $Pt/$anatase $TiO_2$ (via the sol–gel method), and $Pt/$rutile $TiO_2$ (via the sol–gel method), and its CO oxidation performance was significantly affected by the $TiO_2$ crystal phase. The results show that the formation of electron-rich Pt with lower Pt-Pt and Pt-O coordination numbers on the surface of the rutile $TiO_2$ synthesized via the hydrothermal method played a key role. For the study of the $TiO_2$ crystalline phase, more emphasis is placed on the study of single anatase $TiO_2$ or single rutile $TiO_2$ as a support, but the effects of the anatase phase and rutile $TiO_2$ in conventional oxidation reactions when different mixed-crystalline-phase $TiO_2$ is used as a support are rarely reported.

In this study, $TiO_2$ precursors were synthesized via the sol–gel method, and a series of continuous crystal phase transition $TiO_2$ was synthesized at different calcination temperatures (single crystal phase and various mixed crystal phases). The effect of different $Pt/TiO_2$ crystal phases on the catalytic oxidation activity of benzene was investigated. In addition, by combining benzene oxidation data with characterization results, the manner by which the $TiO_2$ crystal phase affects the state of Pt loading and the relationship with benzene catalytic oxidation activity was determined. It is worth noting that the SMSI between Pt and different crystal phase $TiO_2$ and the synergy between anatase and rutile are key to affecting catalytic oxidation activity. Overall, this study provides insight into the catalytic oxidation of benzene via different crystalline phase $Pt/TiO_2$ catalysts.

## 2. Results and Discussion

### 2.1. Structural Properties of the Catalysts

Figure 1a shows the XRD spectra of all $Pt/TiO_2$ catalysts. The main diffraction peak for the $Pt/TiO_2$ (400 °C) and $Pt/TiO_2$ (500 °C) catalysts are ascribed to the anatase phase. With increasing $TiO_2$ carrier calcination temperatures, the half-peak width of the $Pt/TiO_2$ (500 °C) catalyst decreased, and the peak intensity was significantly increased; these are attributed to the continuous growth of anatase crystals. The structure of the catalyst underwent a phase transition with a continuous increment of the carrier's calcination temperature. The crystal phase of the catalyst was completely transformed into a rutile structure at 800 °C. Numerous studies have also used XRD to confirm the phase transition process of $TiO_2$ during heating [15–17]. The different crystalline phases of the $TiO_2$ support can interact with Pt via different strong metal supports (SMSI), which will, in turn, affect the surface reactive oxygen content of the catalyst and the dispersion of the supported Pt [13,18,19]. The supported catalysts loaded on $TiO_2$ with different crystal phases displayed different catalytic activities [20,21]. No relevant diffraction peaks for Pt species were detected in the XRD spectra of any of the catalysts, and this may possibly be due to the low Pt content or its uniform dispersion on $TiO_2$. The specific surface area of the catalysts recorded in Table 1 shows a decreasing trend at calcination temperatures above 400 °C, which is attributed to the enhanced sinter of the support particles at elevated calcination temperatures.

**Table 1.** Specific surface areas; activities of catalysts; $H_2$-TPR, XPS, and Pt particle size results.

| Catalysts | Specific Surface $(m^2g^{-1})$ | Catalytic Activity | $H_2$-TPR | XPS [a] | | | Pt Particle Size |
|---|---|---|---|---|---|---|---|
| | | $T_{90}$ (°C) | Initial Reduction Temperature (°C) | $Pt^{4+}/Pt^{2+}$ | $O^{2-}/O_2^{-}$ | $Ti_{2p3/2}$ | nm |
| $Pt/TiO_2$ (400 °C) | 82.7 | 261 | 97 | 0 | 12.9 | 458.26 | 1.29 |
| $Pt/TiO_2$ (500 °C) | 20.3 | 255 | 98 | 0 | 18.9 | 458.26 | 1.36 |
| $Pt/TiO_2$ (550 °C) | 24.6 | 222 | 109 | 0.35 | 20.4 | 458.25 | 2.01 |
| $Pt/TiO_2$ (600 °C) | 10.5 | 204 | 126 | 0.54 | 1.75 | 458.24 | 5.91 |
| $Pt/TiO_2$ (700 °C) | 8.3 | 201 | 142 | 1.53 | 2.46 | 458.12 | 2.33 |
| $Pt/TiO_2$ (800 °C) | 8.1 | 202 | 139 | 1.05 | 8.14 | 458.11 | 1.72 |

[a] Calculated via XPS deconvolution results.

Raman spectroscopy plays an important role in the study of supported catalysts. To further evaluate the effects of the different $TiO_2$ crystalline phases on the valence state and morphology of Pt species, Raman spectroscopy was performed, and the spectra are shown in Figure 1b. In general, the Raman bands belonging to the anatase phase are around 149 $cm^{-1}$, 201 $cm^{-1}$, 400 $cm^{-1}$, 519 $cm^{-1}$, and 639 $cm^{-1}$, whereas those belonging to the rutile phase are around 425 $cm^{-1}$ and 610 $cm^{-1}$, respectively. Compared with the $Pt/TiO_2$ (400 °C) catalyst, the $Pt/TiO_2$ (550 °C) and $Pt/TiO_2$ (600 °C) catalysts showed slight redshifts at the peaks around 149 $cm^{-1}$, and these may be attributed to the enhanced interactions between Pt and $TiO_2$ [22]. In addition, the bands at 149 $cm^{-1}$ for the $Pt/TiO_2$ (400 °C) and $Pt/TiO_2$ (550 °C) catalysts are broader than those of other catalysts, and this is attributed to the existence of a large number of defects on the surface of powder particles [5]. Specific defects in each catalyst were identified via electron paramagnetic resonance (EPR) (Figure 1c). With the increasing number of rutile facies in $TiO_2$, the bands around 425 $cm^{-1}$ for $Pt/TiO_2$ (700 °C) and $Pt/TiO_2$ (800 °C) were redshifted by 4 $cm^{-1}$ and 9 $cm^{-1}$, respectively, compared to that of $Pt/TiO_2$ (600 °C). The redshift can be attributed to the formation of a structure similar to àPt-O-Ti, which affected the vibrational frequency of the catalysts.

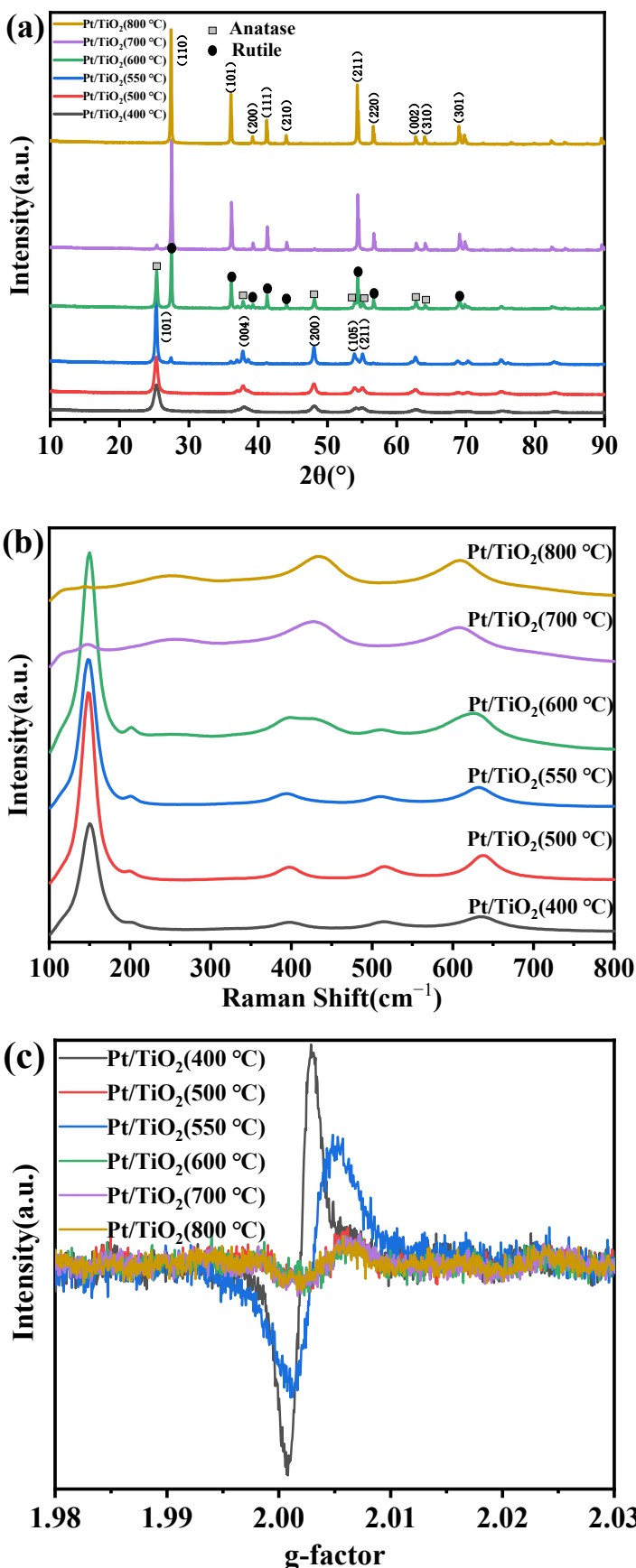

**Figure 1.** XRD patterns of the Pt/TiO$_2$ catalysts (**a**), Raman spectra of the Pt/TiO$_2$ catalysts (**b**), and EPR spectra of individual catalysts obtained at room temperature (**c**).

## 2.2. Benzene Catalytic Performance

Figure 2a shows the relationship between the conversion rate and the reaction temperature of the $C_6H_6$ catalytic oxidation of $Pt/TiO_2$ catalysts. It is obvious that the conversion rate of $C_6H_6$ in the catalyst exhibited a volcanic trend with increasing reaction temperature. The catalytic activity followed the following sequence: $Pt/TiO_2$ (400 °C) < $Pt/TiO_2$ (500 °C) < $Pt/TiO_2$ (550 °C) < $Pt/TiO_2$ (600 °C) < $Pt/TiO_2$ (700 °C) ≈ $Pt/TiO_2$ (800 °C). Moreover, their corresponding $T_{90}$ values were 261 °C, 255 °C, 222 °C, 204 °C, 201 °C, and 202 °C, respectively. Among them, $Pt/TiO_2$ (600 °C), $Pt/TiO_2$ (700 °C), and $Pt/TiO_2$ (800 °C) exhibited similar catalytic oxidation performances as benzene. Figure 2b shows an enlarged view of the activity curves of the catalysts at low temperatures, and it is clear that the optimum catalytic activities were exhibited by the $Pt/TiO_2$ (700 °C) and $Pt/TiO_2$ (800 °C) catalysts. We believe that the crystal phase affects the surface properties of $TiO_2$ and, thus, the catalytic activity of the catalyst. According to the data presented in Table 2 [6,8,23], our $Pt/TiO_2$ (700 °C) catalyst shows the best oxidation benzene oxidation performance. In addition, its crystal form is dominated by the anatase phase [23], which is similar to the crystal phase composition of the $Pt/TiO_2$ (550 °C) catalyst in this study, and it shows similar catalytic activity under the same conditions. This is in agreement with our experimental results. However, the activity of $Pt/TiO_2$ catalysts after reducing atmosphere pretreatment was significantly improved, and their superior performance was a result of the enhancement of SMSI at the Pt-$TiO_2$ interface. This study focuses on the effects of $TiO_2$ on the Pt state and benzene oxidation activity during continuous crystal phase transitions. This research is therefore necessary in order to further improve catalyst performances in this area.

**(a)**

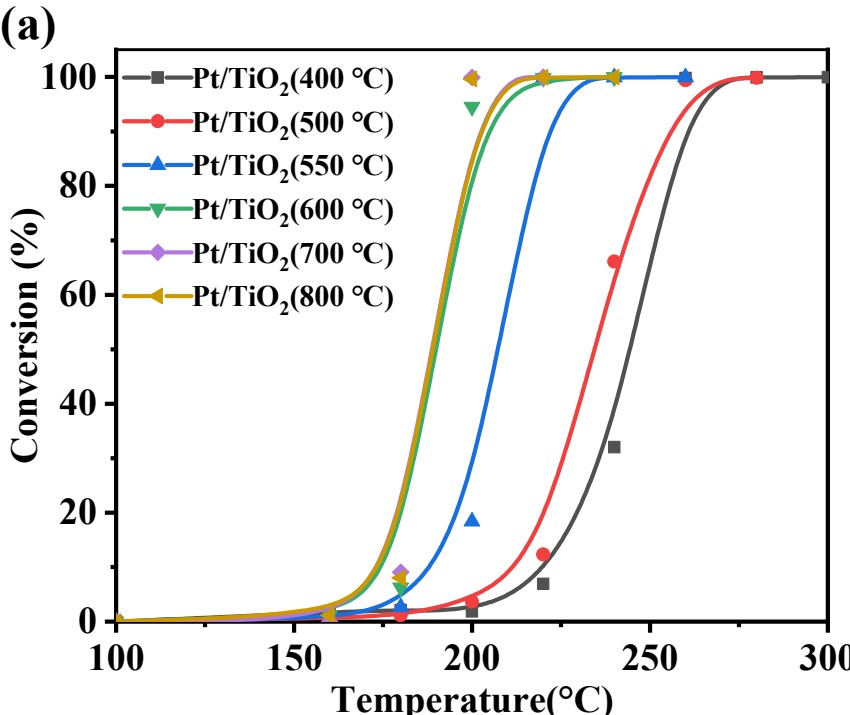

**Figure 2.** *Cont.*

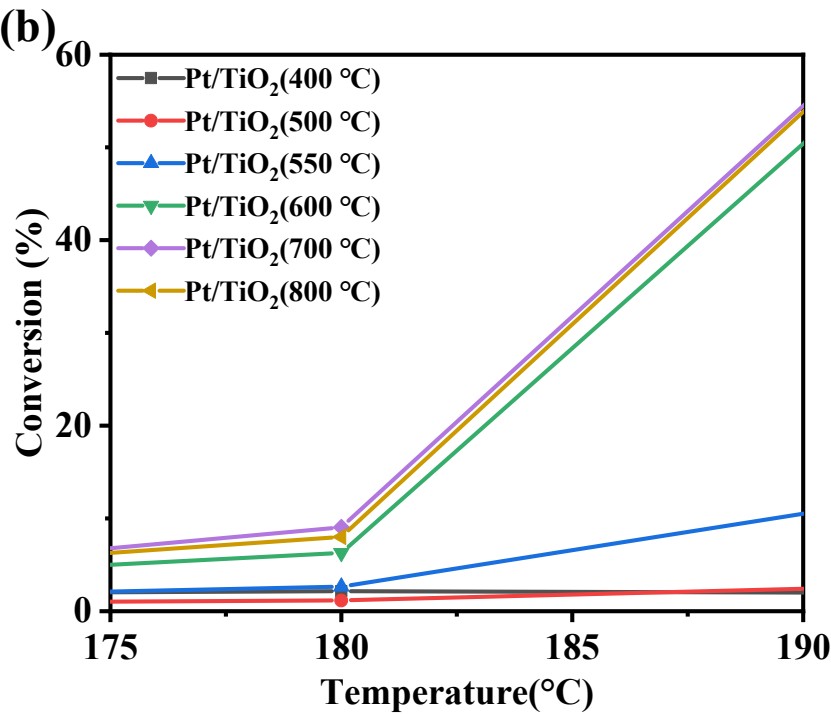

**Figure 2.** Relationship between the benzene conversion rate and reaction temperature of different crystal forms of Pt/TiO$_2$ (**a**) and the low-temperature performance curve of the catalytic combustion of benzene with different crystal forms of Pt/TiO$_2$ (**b**).

**Table 2.** Comparison of the thermal catalytic oxidation of gaseous benzene over supported platinum catalysts.

| Catalyst | Catalyst Activation | Benzene Concentration (ppm) | Space Velocity | T$_{90}$ (°C) | Reference |
|---|---|---|---|---|---|
| 1% Pt/TiO$_2$ | H$_2$ pretreatment | 100 | 8696 mLg$^{-1}$h$^{-1}$ | 167 | [6] |
| 1% Pt/TiO$_2$ | Impregnation method | 1000 | 60,000 mLg$^{-1}$h$^{-1}$ | >225 | [23] |
| 1% Pt/TiO$_2$ | H$_2$ pretreatment and Impregnation method | 1000 | 60,000 mLg$^{-1}$h$^{-1}$ | 180 | [23] |
| 1% Pt/TMS | H$_2$ pretreatment | 400 | 60,000 mLg$^{-1}$h$^{-1}$ | 178 | [8] |
| 1% Pt/TiO$_2$ (700 °C) | Impregnation method | 1000 | 60,000 mLg$^{-1}$h$^{-1}$ | 201 | This work |

*2.3. Analysis of Catalyst Morphology and Interaction Mechanisms*

The surface morphology of each catalyst was further investigated via scanning electron microscopy (SEM), and as shown in Figure 3a–f, the micrographs reveal the morphological characteristics of the catalyst support at different calcination temperatures. For the Pt/TiO$_2$ (400 °C) and Pt/TiO$_2$ (500 °C) catalysts, flat spherical particles with rough surfaces were formed via the accumulation of many small particles. The formation of trace amounts of smooth surfaces on the catalysts was observed when the calcination temperature of the TiO$_2$ support was increased to 550 °C, and this is attributed to the high-temperature sintering applied on large crystals, which is consistent with the BET test results. The morphology gradually changed to a smooth cube as the calcination temperature of the support increased. From the specific surface area results of each sample in Table 1, it can be seen that the specific surface area of the catalyst tends to decrease with the increase of the calcination temperature of the TiO$_2$ support. Under the same Pt loading, the specific surface area of the support has a certain influence on the dispersion of Pt species. A high specific surface area is more beneficial to the dispersion of Pt species, while a low specific surface area is

more likely to cause aggregation of Pt species. Therefore, we further analyzed each catalyst using TEM. Figure 4a–f shows the TEM images of different Pt/TiO$_2$ catalysts. The different crystalline phases of the catalyst produced Pt nanoparticles of different sizes. In the case where the TiO$_2$ support was a pure anatase crystal, Pt existed only as a nanoparticle with an average particle size of 1.36 nm. With the onset of rutile phase formation, the particle size of Pt first increased and then decreased with the phase transition of the TiO$_2$ support. On the other hand, when the catalyst possessed a pure anatase structure, Pt did not exhibit a specific crystal facet; in contrast, on the surface of the Pt/TiO$_2$ (550 °C) catalyst, Pt existed as nanoparticles exhibiting a specific Pt(111) crystal facet. However, when the Pt/TiO$_2$ (600 °C) catalyst was composed of mixed phases in similar proportion, Pt agglomerated with an average particle size of 5.91 nm, which was the maximum particle size measured. It was believed that this was due to agglomeration caused by the roughness of the catalyst's surface potential due to lattice disorder during the transition of the crystal form. Drushan et al. [24] reported improved redox properties with TiO$_2$ lattice disorder at the anatase–rutile transition. Based on the analysis of TEM and XRD data, the size of Pt species on the catalyst gradually increases with the decrease in specific surface area when the calcination temperature of the support is below 600 °C. However, when the calcination temperature of the support is above 600 °C (i.e., when it is converted to rutile phase), the morphology of Pt species undergoes re-dispersal rather than further aggregation with the decrease of specific surface area, indicating that the crystal form of the catalyst plays a dominant role in the morphology of Pt species. The excellent catalytic activity of the Pt/TiO$_2$ (600 °C) catalyst can be explained by the fact that the larger Pt nanoparticle sizes impeded its bonding with oxygen, which affected the formation of the inert PtO$_2$ species, thereby increasing both the number of active sites for O$_2$ chemical adsorption and the rate of oxidation reaction [25,26]. As shown in Figure 5a, three types of oxygen species were detected in the O1s orbital of the Pt/TiO$_2$ (600 °C) catalyst, with peaks at around 529, 531, and 533 eV, corresponding to surface lattice oxygen (O$^{2-}$), surface oxygen (O$^{2-}$), and surface hydroxyl oxygen (-OH), respectively [27–29]. The Pt/TiO$_2$ (600 °C) catalyst exhibited a lower O$^{2-}$/O$_2{}^-$ ratio, which further proves that the activity of the catalyst is closely related to the active oxygen species content on its surface. With the substantial rutile phase in the TiO$_2$ support, the Pt of the Pt/TiO$_2$ (700 °C) and Pt/TiO$_2$ (800 °C) catalysts was redispersed into flat spherical nanoparticles, and the particles at the edge of the catalyst displayed more typical Pt(111) crystal facets, indicating that the wet surface of the rutile TiO$_2$ was more beneficial for SMSI between Pt and TiO$_2$ [30–33]. Using DFT calculations, Zhang et al. [13] showed that the diffusion barrier of Pt on rutile TiO$_2$ was much higher than that on anatase TiO$_2$, and the stronger interactions and higher diffusion barrier favored the confinement of Pt to a specific position. The results show that the different crystal phases of TiO$_2$ exhibit different effects on the dispersion and particle size of Pt. The formation of Pt(111) crystal planes in the catalyst benefitted the dissociation and adsorption of O$_2$ molecules adsorbed on the catalyst surface [34,35]. Employing scanning tunneling microscopy images, Cerhard Ertl [36] explained that the O$_2$ molecules adsorbed on the crystal plane of Pt(111) formed tricoordinated oxygen atoms after dissociation, which significantly reduces the activation energy of oxidation reactions. As it is known, benzene molecules are VOC molecules containing $\pi$ bonds, which also undergo strong chemical adsorption on the surface of Pt species to interact with the oxygen atoms adsorbed on the surface. This means that the presence of more adsorbed oxygen species on the surface of Pt species is advantageous to the catalytic oxidation of benzene.

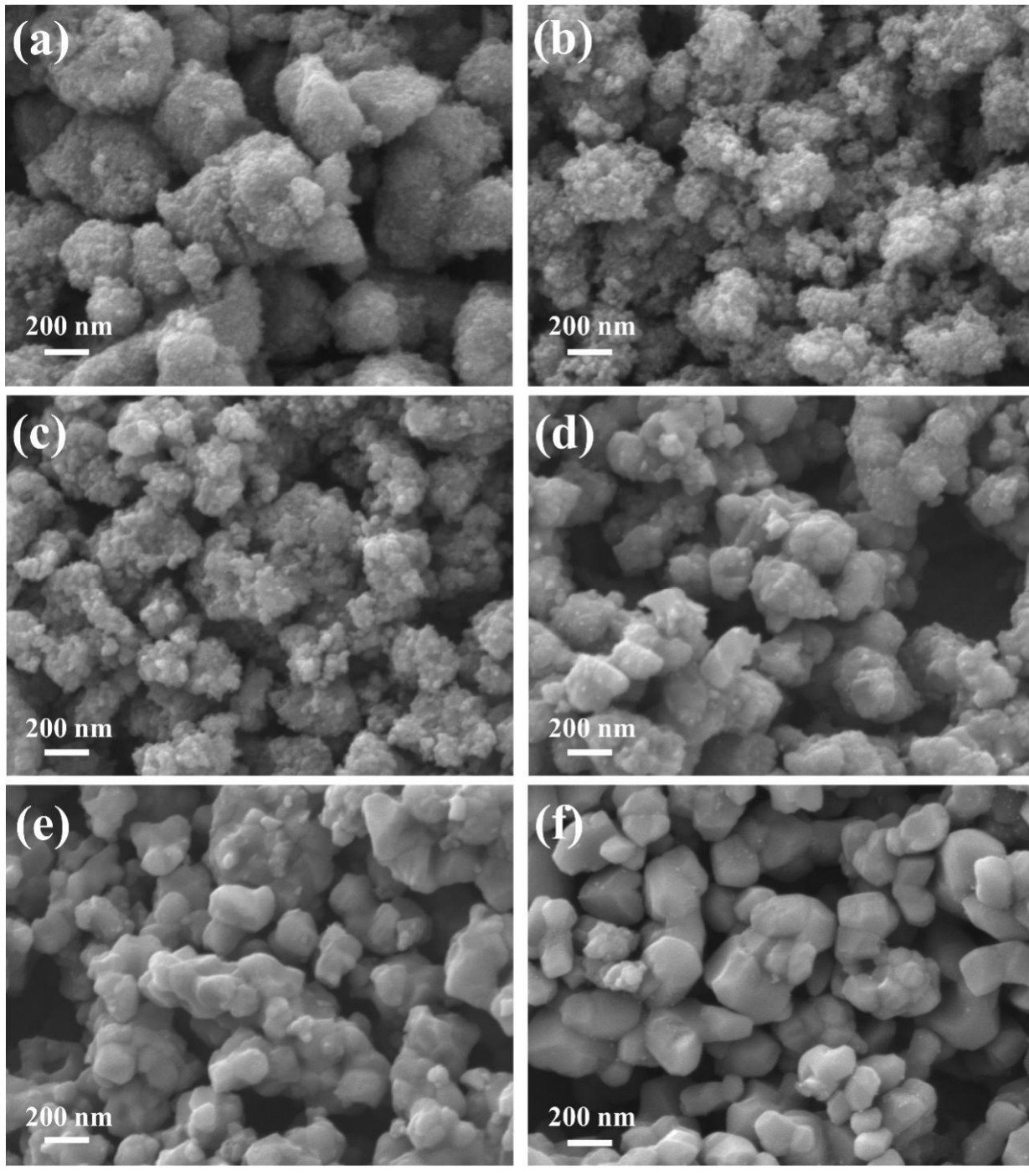

**Figure 3.** SEM photographs of the Pt/TiO$_2$ catalyst: (**a**) Pt/TiO$_2$ (400 °C); (**b**) Pt/TiO$_2$ (500 °C); (**c**) Pt/TiO$_2$ (550 °C); (**d**) Pt/TiO$_2$ (600 °C); (**e**) Pt/TiO$_2$ (700 °C); (**f**) Pt/TiO$_2$ (800 °C).

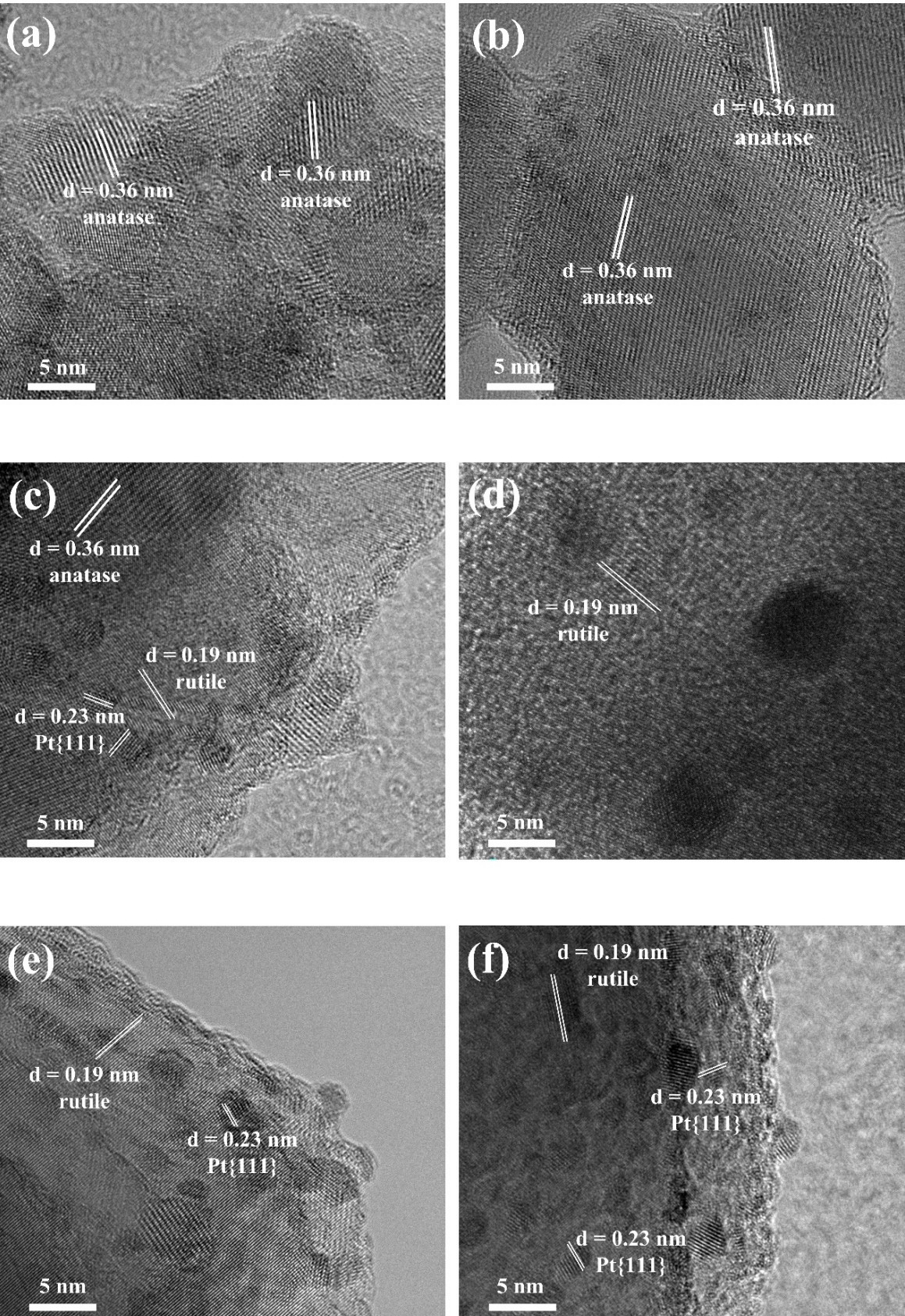

**Figure 4.** TEM photographs of the Pt/TiO$_2$ catalyst: (**a**) Pt/TiO$_2$ (400 °C); (**b**) Pt/TiO$_2$ (500 °C); (**c**) Pt/TiO$_2$ (550 °C); (**d**) Pt/TiO$_2$ (600 °C); (**e**) Pt/TiO$_2$ (700 °C); (**f**) Pt/TiO$_2$ (800 °C).

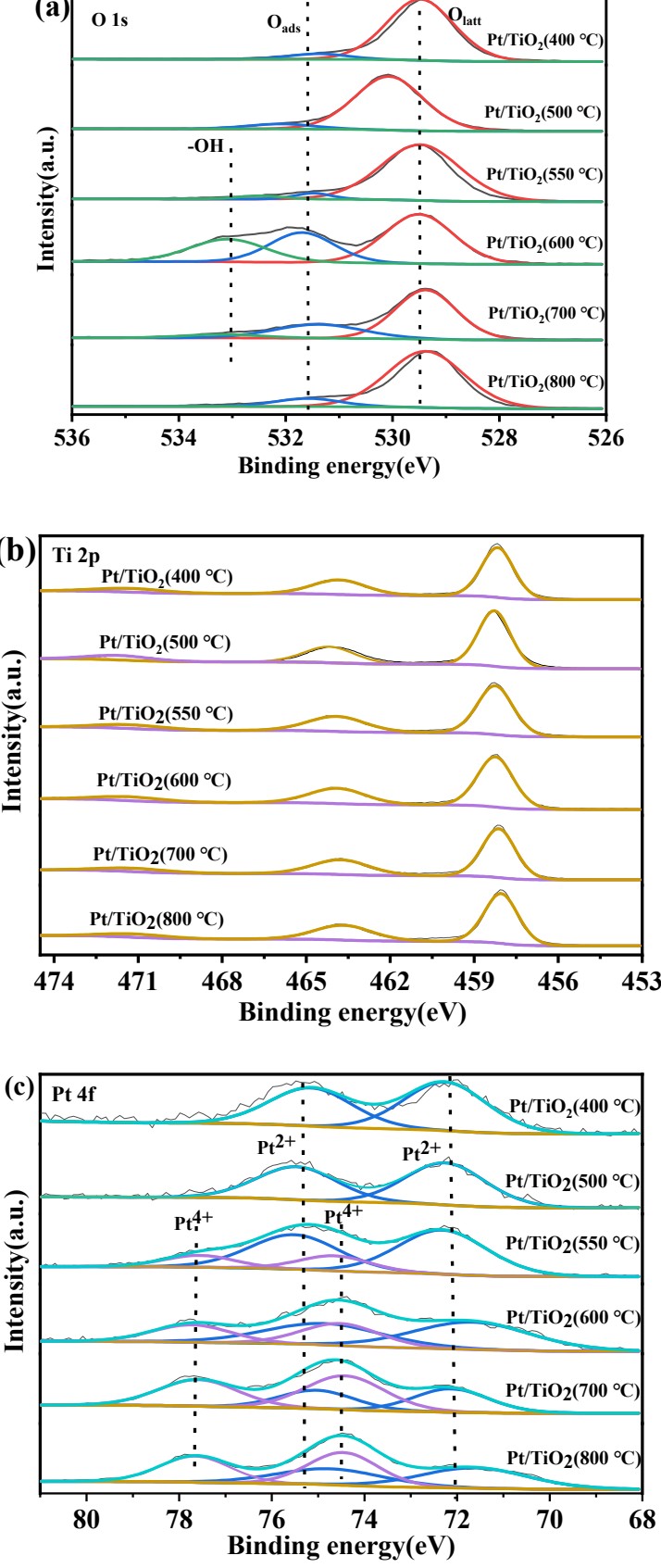

**Figure 5.** $O_{1s}$ (**a**), $Ti_{2p}$ (**b**), and $Pt_{4f}$ (**c**). XPS spectra of $Pt/TiO_2$ catalysts.

As shown in Figure 5c, the XPS spectra of $Pt_{4f}$ photoelectrons are presented in order to elucidate the different oxidation states of the Pt metal deposited on the $TiO_2$ support and the surface chemical states of the elements. The $Pt_{4f7/2}$ and $Pt_{4f5/2}$ signals around 72.0 and 75.2 eV are attributed to $Pt^{2+}$, while the $Pt_{4f7/2}$ and $Pt_{4f5/2}$ signals around 74.6 and 77.8 eV are attributed to $Pt^{4+}$. It is worth noting that when the $TiO_2$ crystal is characterized by pure anatase, the Pt in the $Pt/TiO_2$ (400 °C) and $Pt/TiO_2$ (500 °C) catalysts chiefly existed in the form of $Pt^{2+}$; moreover, at the onset of the $TiO_2$ phase transition, the Pt on the surface of the catalyst started to appear in a mixed valence state of $Pt^{2+}$ and $Pt^{4+}$. This can be explained by the different crystal structures and energy bands of $TiO_2$ in the mixed crystal. The energy bands of both phases overlapped, and the mutual transfer of electrons and holes between the two phases promoted $Pt^{2+}$ to $Pt^{4+}$ conversion [37–39]. The $Pt^{4+}/Pt^{2+}$ values of all catalysts are shown in Table 1. With increasing $Pt^{4+}$ content, the activity of the catalyst first increased significantly and then remained unchanged, indicating that the increase in activity was not caused by all the constituent $Pt^{4+}$. Consequently, a small number of $Pt^{4+}$ formed isolated $PtO_2$ species that did not interact with $TiO_2$, and a single $PtO_2$ species was inert and had very low catalyst activity [40]. Hence, activity was influenced by the interaction between the highly oxidized Pt species and $TiO_2$ at the edge of the interface with Pt particles. As shown in Figure 5b, all samples displayed two peaks at 463.8–464 eV and 458–458.4 eV, belonging to the $Ti_{2p1/2}$ signal of $Ti^{4+}$ and the $Ti_{2p3/2}$ signal, respectively. The binding energy of $Ti_{2p3/2}$ decreased slightly with the transition from anatase $TiO_2$ to rutile $TiO_2$, and the resultant lower binding energy indicates that $Ti^{3+}$ was present on the surface of the catalyst. It has been established that lower energy Ti atoms have electron vacancies that facilitate the acquisition of more mobile electrons that can promote the formation of $Pt^{2+}/Pt^{4+}$ [41–43]. The formation of SMSI is closely related to the bonding between the Pt species and the cationic or atomic Ti species [44]. A strong interaction occurred between $Pt^{4+}$, a highly oxidized $Pt^{4+}$ at the edge of the Pt particle, and the low energy atom Ti, forming a structure similar to àPt-O-Ti [45], which enhanced charge transfers with oxygen species via interactions between the Ti species and highly oxidized Pt species. The àPt-O-Ti species is more prone to oxidation reactions, which is one of the reasons for the improved oxidation performances of the catalysts.

In order to investigate the redox properties of $Pt/TiO_2$ catalysts, $H_2$-TPR experiments were carried out. As shown in Figure 6a, the reduction peak occurring at around 50 °C–150 °C is attributed to the reduction of $Pt_{OX}$ to $Pt^0$ ($Pt^{2+}$ and $Pt^{4+} \rightarrow Pt^0$). The shift in reducing peaks belonging to the Pt species to higher temperatures indicates an enhanced interaction between surface-dispersed Pt species and $TiO_2$. The two reduction peaks within the range of 250 °C to 600 °C belong to surface-adsorbed oxygen and subsurface-adsorbed oxygen, respectively [46]. With the transition of $TiO_2$ to the rutile phase, the relative content of adsorbed oxygen on the surface of the catalyst increases, and its reduction temperature gradually decreases; moreover, the lower reduction temperature may be beneficial to improving the catalytic oxidation performance of the catalyst, especially at low temperatures. This result is also consistent with the low-temperature oxidation activity of benzene.

To further investigate the properties of the Pt species on the different catalysts, CO-FTIR was used to distinguish the presence of the Pt species, and the spectra are illustrated in Figure 6b. It has been reported that the asymmetric band observed around 2050~2060 cm$^{-1}$ is due to the linear adsorption of CO in the low coordination Pt edge and Pt corner, and the adsorption band at 1850 cm$^{-1}$ belongs to the bridge adsorption of CO in Pt, which is characteristic of nanoscale Pt particles [19,47]. Based on the XPS and Raman analysis described above, a weak peak of CO adsorption was observed around 2098 cm$^{-1}$, which is attributed to the characteristic peak of CO adsorption on a highly oxidized $Pt_1$ atom at the edge of the Pt particle. The characteristic peak of linear CO adsorption on Pt oxide species ($Pt_{OX}$) was detected in the band around 2120 cm$^{-1}$ [48]. Furthermore, the infrared CO adsorption results are in agreement with the above analytical results.

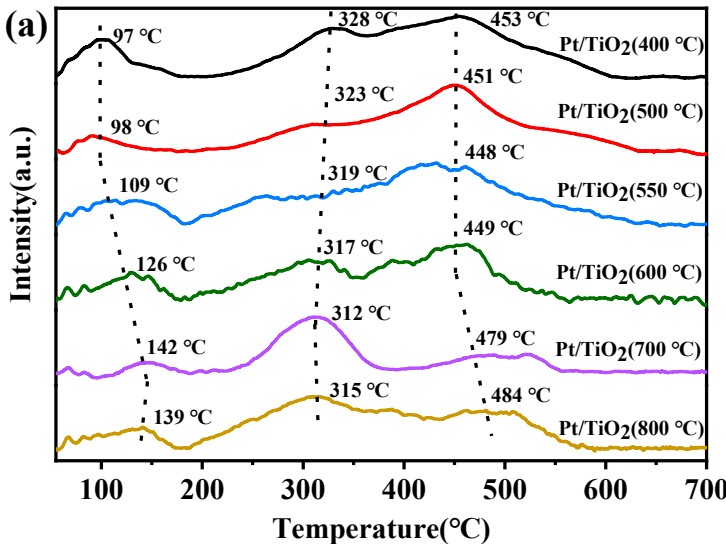

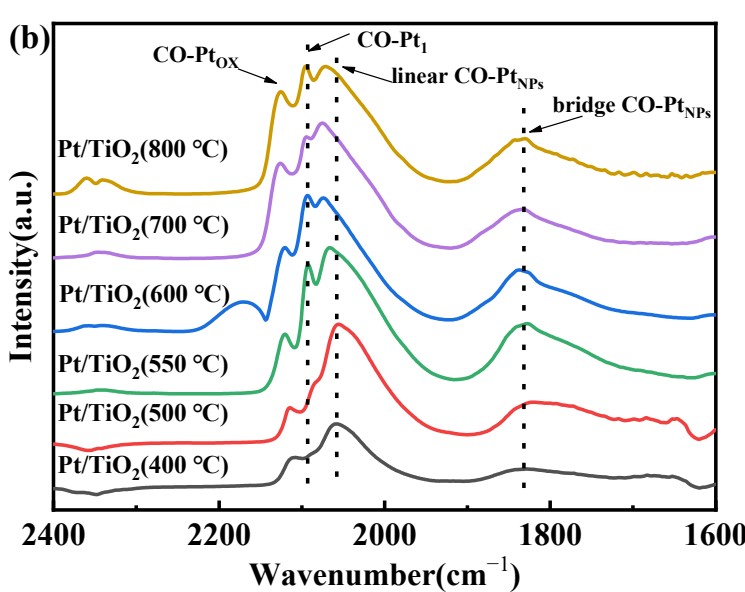

**Figure 6.** $H_2$-TPR profile of Pt/TiO$_2$ (**a**) and FTIR spectra of CO adsorption on Pt/TiO$_2$ (**b**).

## 3. Experimental Section

### 3.1. Materials

The tetrabutyl titanate (C$_{16}$H$_{36}$O$_4$Ti, ≥99.0%), acetylacetonate (C$_5$H$_8$O$_2$, 99.0%), and platinum nitrate solution (PtN$_2$O$_6$, Pt, 18.02%) used in this work were purchased from Shanghai Aladdin Biochemical Technology Co., Ltd. (Shanghai, China). Hydrochloric acid solution (HCl, 37.5%) and absolute ethanol (C$_2$H$_6$O, 99.8%) were purchased from the Sinopharm Chemical Reagent Company (Shanghai, China). The reaction mixture of the bottle was 1000 ppm benzene + 4% O$_2$ + balanced nitrogen, which was provided by Shanxi Yihong Gas Industry Co., Ltd. (Linfen, China). All chemicals were used as received.

### 3.2. Preparation of Catalysts

#### 3.2.1. Preparation of TiO$_2$

TiO$_2$ supports were prepared via the sol–gel method, and the detailed preparation procedure is described as follows: 15 mL of tetrabutyl titanate and 1 mL of acetylacetone were added to 120 mL of absolute ethanol, and they were mixed properly, followed by the

dropwise addition of 30 mL of deionized water to the mixture while stirring. The resultant solution was refluxed at 80 °C for 3 h, and the product was naturally cooled, collected via high-speed centrifugation, thoroughly washed with absolute ethanol, and dried at 80 °C. Finally, the dried solid was ground into powder and heated to 400 °C, 500 °C, 550 °C, 600 °C, 700 °C, and 800 °C at a rate of 5 °C/min. The $TiO_2$ products were obtained after calcination for 5 h, and they were labeled $TiO_2$ (400 °C), $TiO_2$ (500 °C), $TiO_2$ (550 °C), $TiO_2$ (550 °C), $TiO_2$ (600 °C), $TiO_2$ (700 °C), and $TiO_2$ (800 °C).

### 3.2.2. Preparation of $Pt/TiO_2$

Using the equal volume impregnation method to load a mass fraction of 1 wt% Pt, it was first necessary to measure the water absorption rate of the $TiO_2$ carrier synthesized under different conditions. After determining the water absorption rate of the carrier, a certain amount of platinum nitrate solution was dissolved in deionized water and sonicated for 10 min to dissolve it uniformly. Then, a certain amount of 40–60 mesh $TiO_2$ carrier was weighed, impregnated in platinum nitrate solution, and allowed to stand for 10 min. The temperature was increased from room temperature to 110 °C at a heating rate of 5 °C/min, and drying occurred for 12 h. The $Pt/TiO_2$ catalyst was obtained via calcination in air from room temperature to 400 °C for 5 h.

### 3.3. Catalyst Characterization

X-ray diffraction (XRD) analysis was carried out using Cu-Ka radiation on an RM-600 X-ray powder diffractometer (Tokyo Japan). A Renishaw (Dundee, IL, USA) inVia Raman spectrometer was used to detect laser Raman spectra under excitation at 532 nm. $N_2$ adsorption–desorption measurements were conducted on a Micromeritics ASAP 2460 (Norcross, GA, USA). Transmission electron microscopy (TEM) images were obtained in the energy spectrum signal of JED-2300T(Tokyo Japan). Scanning electron microscopy (SEM) images were obtained using JSM-7900F (Tokyo Japan) at 5 kV. The surface chemical state of the sample was analyzed via X-ray photoelectron spectroscopy (XPS) (Waltham, MA, USA), and the binding energies of all elements were corrected against the standard C1s line of 284.6 eV. Electron paramagnetic resonance (EPR) was carried out using a Burker (Hurstville, NSW, Australia) EMX Plus electron paramagnetic spectrometer. A $H_2$ temperature-programmed reduction ($H_2$-TPR) experiment was performed using a Micromeritics (Norcross, GA, USA) Auto Chem II 2920 chemisorption instrument. For the $H_2$-TPR experiment, 40 mg of the catalyst was placed in a quartz tube. After cooling the catalyst from 200 °C to room temperature, a 10 vol%$H_2$/Ar mixture was introduced into the tube reactor at a flow rate of 20 mL/min. Finally, the catalyst was heated to 800 °C at a heating rate of 10 °C/min. Infrared CO adsorption was measured using a Nicolet (Nicolet, QC, Canada) 6700 infrared spectrometer.

### 3.4. Catalytic Evaluation

The catalytic evaluation was carried out in a tubular fixed-bed reactor. Typically, 200 mg of the catalyst (40–60 mesh) was loaded into a quartz tube. Then, 1000 ppmV benzene, 4 vol% $O_2$, and $N_2$ were introduced into the reactor as equilibrium gases at a total flow rate of 200 mL/min, corresponding to a weight hourly flow rate (WHSV) of 60,000 mL/(g·h). The benzene conversion was calculated using the following formula:

$$X(\%) = \frac{C_{in} - C_{out}}{C_{in}} \times 100\%$$

where X denotes the benzene conversion, and $C_{in}$ and $C_{out}$ denote the inlet and outlet concentrations of benzene that were measured using online gas chromatography methods with FID detectors.

## 4. Conclusions

In summary, $TiO_2$ precursors were synthesized via the sol–gel method, and different crystal phases of $TiO_2$ were obtained by adjusting the calcination temperature of the precursors, followed by an investigation of the catalytic oxidation of benzene after Pt loading. The strong interactions between the different crystal phases of $TiO_2$ and Pt played key roles in the catalytic oxidation of benzene. The rutile phase is more beneficial to the formation of specific Pt(111) crystal faces relative to the adsorption of $O_2$ molecules and the dissociation of O atoms into tricoordinates. The catalytic oxidation of benzene molecules is closely related to the formation of àPt-O-Ti species between $Pt^{4+}$ and $TiO_2$ at the edge of the Pt nanoparticles. Benzene molecules were adsorbed on both the catalyst's surface and the active oxygen near $Pt^{4+}$ at the edge of the Pt particles. Consequently, the benzene molecules were oxidized as target products, the active oxygen species participated in the reaction, the $Pt^{4+}$ on the catalyst surface was reduced to a lower valence state, and the gas phase oxygen oxidized the lower valence Pt. Based on the results from TEM, Raman, XPS, and $H_2$-TPR, varying interactions between $TiO_2$ and Pt were observed for the different crystal phases, and the Ti atoms with lower binding energy, as well as the electron transfer between the anatase and rutile phases, further promoted the conversion of $Pt^{2+}$ to $Pt^{4+}$, which justifies the observed higher $Pt^{4+}/Pt^{2+}$ ratio of the $Pt/TiO_2$ (700 °C) catalyst. Catalytic activity was predominantly influenced by the àPt-O-Ti species formed between $Pt^{4+}$ and Ti at the edge of the Pt nanoparticles, which promoted the flow of electrons and the transfer of reactive oxygen species, thereby exhibiting the best catalytic activity.

**Author Contributions:** H.C.: Conceptualization; data curation; formal analysis; writing—original draft. J.H.: Conceptualization; formal analysis; project administration; writing—original draft; visualization. D.W.: Writing—review and editing; data curation; resources. W.B.: Funding acquisition; data curation. C.H.: Writing—review and editing; data curation; resources. X.L.: writing—review and editing; data curation; resources. L.C.: Data curation; resources; visualization. J.W.: Funding acquisition; data curation. All authors have read and agreed to the published version of the manuscript.

**Funding:** This work was funded by the Natural Science Research Project of Shanxi Basic Research Program (202203021221064), the Program of Shanxi Zhejiang University New Materials and Chemical Research Institute (2021SX-TD007), the Shanxi Province Science and Technology Innovation Talent Team (202204051002025), and the Environmental Catalytic Materials Technology Innovation Center of Shanxi Province (202104010911008).

**Data Availability Statement:** The data that support the findings of this study are available upon reasonable request from the authors.

**Conflicts of Interest:** The authors declare that they have no known competing financial interests or personal relationships that could have appeared to influence the work reported in this paper.

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
