# Peer review of "Effect of Interaction between Pt and Different Crystalline Phases of TiO2 on Benzene Oxidation"

_catalysts, doi:10.3390/catal14040234_

Round 1
Reviewer 1 Report
Comments and Suggestions for Authors
Scientific comments:
Line 66: Main crystalline phase of titania are Rutile, Anatase and brookite while another name is mentioned instead of brookite in the text
Can author include the FTIR spectra of samples in wider region (400-3800 cm-1). This way readers can see if Ti-O-Pt species have vibrational band in IR spectra or not?
TEM image alone is not a strong proof for existence or absence of Pt{111} facet. Do you have any characterization data to proof that {111} facet of Pt preferably form on Rutile support?
My point of view: Anatase samples have much higher surface area than Rutile samples and therefor to have same surface density, more Pt should loaded on anatase containing samples depending on their surface area. This is necessary to have similar condition for comparison of the samples. In this manuscript similar to many other, same amount of Pt is loaded on the samples with different surface area which increase the chance of the formation of larger particle on the low surface area samples (i.e. Rutile).
How reproducible are the location of reduction peak in the recorded TPR data? Why samples with same phases have different reduction peak and the change is irregular ( samples prepared at 600°C, 700°C and 800 °C)
What about other phases of TiO2? Did you tried to investigate brookite phase as well?
Loading percentage of Pt is not mentioned in the abstract and conclusion. Also, the selected 1% loading of Pt is higher than 0.5% Pt loading reported in some of the literature (e.g. Chemosphere Volume 265, =2021, 129127).
Comments on the Quality of English Language
General comments:
There are numerous language errors or sentences with unclear meaning. The manuscript need proofread to address these errors. Example of sentences with unclear meaning:
1-First sentences of the abstract should be rephrased and has wrong meaning. Benzene is not oxidation catalyst. TiO2-supported Pt catalysts is the catalyst for oxidation (see following part of the sentence): “on the catalytic oxidation properties of benzene for TiO2-supported Pt catalysts”
2-Pt-O-Ti structure à Pt-O-Ti species
3-Line 29: the abbreviation PM is used without any description. All abbreviation should be explained in their fist appearance
3-Line38 and 39: the sentence should be rephrased. The claim “the main catalyst types are the supported noble metal catalysts” is not true and misleading (probably authors means for oxidation reaction)
4-Line 50: “Recently, the benzene oxidation study with Pt as the active metal accompanied by the reducible oxide TiO2 was carried out under certain conditions, according to the research of Liu et al”
5-Line 92:” continuous crystal phase transition TiO2 were synthesized”
6-replace roasting with calcination in the text (roasting is not a common word in this case)
7-Line 54: “were more conducive to the dispersion of Pt”
8-Fist line of conclusion :” TiO2 precursors of were synthesized by the sol-gel method”
Author Response
Please see the attachment. English grammar issues have been corrected in the text.

Reviewer 2 Report
Comments and Suggestions for Authors
The article focuses on evaluating the impact of different crystalline phases of titanium dioxide (TiO2) on the catalytic oxidation properties of benzene in the presence of TiO2-supported platinum (Pt) catalysts. The study involves various spectroscopic and analytical techniques to examine the physicochemical properties of the catalysts. The findings indicate that the different TiO2 crystalline phases have noticeable effects on the valence state and morphology of Pt. Specifically, the rutile phase is highlighted for favoring the formation of specific Pt(111) crystal faces, which, in turn, enhances the amount of surface-active oxygen species. Additionally, the formation of a Pt-O-Ti structure between Pt4+ and Ti at the edge of Pt nanoparticles is observed. This structure promotes electron flow and the transfer of reactive oxygen species, contributing to the overall catalytic activity in the oxidation of benzene. I recommend the article be published after the following minor corrections are attended to:
1. Proofreading is recommended as the are minor gramatical errors e.g. the is some redundancy in lines 126 – 128
2. Fig 1 b. is not a true reflection of Fig 1a in the zoomed area. e.g. the conversion for catalyst Pt/TiO2(600), Pt/TiO2(700) and Pt/TiO2(800) is more that 80% at temperature of 190 in Figure 1 a but it is shown to be less that 60% in Fig. 1b
3. Line 173 what isT90 and how is it related to activity
4. Line 177… it is claimed that only catalyst Pt/TiO2(700) and Pt/TiO2(800) are optimum and explanation of why Pt/TiO2(600) is excluded is not given while there is no much difference between the activity of the three catalyst as shown in figure 1a. in fact their T90 values are similar
5. The comparison made in line 177 is not quite clear. Please rephrase for clarity
6. Line 183, a statement highlighting that activity is increased by reducing conditions is given, but such results are not given
7. The anatase phase seems to reduce the activity only upto a certain ratio, below which (Tcal > 600), a Rietvel refinement is recommended to obtain such threshold in terms of phase composition.
8. According XRD results in figure 2a, a mixture of anatase and rutile starts forming at a calcination temperature of 550 degrees Celsius, but Raman results in Figure 2b, do not indicate any mixture of phases at 550? Why is this?
9. Raman results needs some re-interpretation see following work for better interpretation:
a) Maftei, A.E. et al. “Micro-Raman—A tool for the heavy mineral analysis of Gold Placer-Type Deposits (Pianu Valley, Romania)”, Minerals 10 (2020) 988
b) Ekoi E. et al., “Characterization of titanium dioxide layers using Raman spectroscopy and optical profilometry: Influence of oxide properties”, Results Phys. 14 (2019) 1574. Interpretation of Fig. 2C is missing.
10. Figure 4, relabel from “a” and so on…. Not from “g”
11. Usually the activity decreases with decreasing surface area but converse is seen in this work, why is this so?
12. The observed trend in O2-/O2- ratio does not match the corresponding trend in the activity of the catalyst… why?
13. In Line 311, a claim is made that Ti3+ existed on the surface. However, the XPS fitting do not show the co-existence of Ti3+ and Ti4+.
14. CO-FTIR analysis is not described anywhere in the paper
Author Response
Please see the attachment. The English language issue has been addressed in the text.

Reviewer 3 Report
Comments and Suggestions for Authors
The manuscript presents an intriguing study on the effect of various TiO2 crystalline phases for Pt catalysts supported on TiO2 on the catalytic oxidation of benzene. a series of physicochemical investigations spectroscopic and analytical methods were carried out to understand the Pt sites and active oxygen species. The Raman. EPR and XPS analysis revealed the information about the oxygen vacancies. This study is quite interesting for the readers. The following are some suggestions and feedback to enhance the clarity and scientific rigor of the manuscript.
1, In 44, please correct the word “researches” to research.
2, Please calculate particle size from the HR-TEM images and corresponding Histograms.
3, How the particle size is calculated in Table.2?,
4, Please compare the Pt particle size from HR-TEM and CO chemisorption.
5, I found only one reference from 2022, Please cite recent references.
Comments on the Quality of English Language
English corrections are needed to improve the quality.
Author Response

(The authors gave the same response as above.)

Round 2
Reviewer 1 Report
Comments and Suggestions for Authors
With additional minor modification manuscript maybe suitable for publication in catalysts: The revised manuscript does not address some of the reviewer's concern:
1-Main crystalline phase of titania are Rutile, Anatase and brookite while another name is mentioned instead of brookite in the text (page 3 line 4)
2-Authors relate most of the observed changes to TiO2 phase transition however surface area of catalyst is also changed during the phase transition and should not rule out when discussing the results.
3-Loading percentage of Pt is not mentioned in the abstract and conclusion. Also, the selected 1% loading of Pt is higher than 0.5% Pt loading reported in literature (e.g. Chemosphere Volume 265, =2021, 129127).
Comments on the Quality of English LanguageLanguage of the text should be improved. Example of language errors and suggestion for improving the language:
instead of structure use species when mentioning Pt-O-Ti: for instance Pt-O-Ti species was formed on the surface
Do not use conductive instead of beneficial: Conductive means the ability to transmit something, often referring to heat or electricity.
First sentence of the abstract: To evaluate the effects of different TiO2 crystalline phases relative to (on) the catalytic oxidation performance of benzene on Pt-loaded TiO2 catalysts
our Pt/TiO2 (700 °C) catalyst shows the best catalytic performance for the Pt/TiO2 catalyst without pretreatment (the best oxidation benzene oxidation performance)
Author Response
Please see the attachment. The language error has been corrected and highlighted in red font.
